# Regulation of *Arabidopsis* photoreceptor CRY2 by two distinct E3 ubiquitin ligases

Yadi Chen[1,4], Xiaohua Hu[1,4], Siyuan Liu[1], Tiantian Su[2], Hsiaochi Huang[2], Huibo Ren[1], Zhensheng Gao[1], Xu Wang[1,2], Deshu Lin[1], James A. Wohlschlegel [3], Qin Wang [1✉] & Chentao Lin [2]

Cryptochromes (CRYs) are photoreceptors or components of the molecular clock in various evolutionary lineages, and they are commonly regulated by polyubiquitination and proteolysis. Multiple E3 ubiquitin ligases regulate CRYs in animal models, and previous genetics study also suggest existence of multiple E3 ubiquitin ligases for plant CRYs. However, only one E3 ligase, Cul4[COP1/SPAs], has been reported for plant CRYs so far. Here we show that Cul3[LRBs] is the second E3 ligase of CRY2 in *Arabidopsis*. We demonstrate the blue light-specific and CRY-dependent activity of LRBs (Light-Response Bric-a-Brack/Tramtrack/Broad 1, 2 & 3) in blue-light regulation of hypocotyl elongation. LRBs physically interact with photoexcited and phosphorylated CRY2, at the CCE domain of CRY2, to facilitate polyubiquitination and degradation of CRY2 in response to blue light. We propose that Cul4[COP1/SPAs] and Cul3[LRBs] E3 ligases interact with CRY2 via different structure elements to regulate the abundance of CRY2 photoreceptor under different light conditions, facilitating optimal photoresponses of plants grown in nature.

[1] College of Life Sciences, Basic Forestry and Proteomics Research Center, Fujian Agriculture and Forestry University, Fuzhou, China. [2] Department of Molecular, Cell & Developmental Biology, University of California, Los Angeles, CA, USA. [3] Department of Biological Chemistry, University of California, Los Angeles, CA, USA. [4] These authors contributed equally: Yadi Chen, Xiaohua Hu. ✉email: qinwangCRY@163.com

CRYs are photolyase-like flavoproteins that act as photo-receptors in plants or core components of the molecular clock in mammals[1–3]. Regulation of CRY abundance is an important mechanism to control cellular photo-responses and chrono-responses. For example, two cullin 1 family E3 ubiquitin ligases, SCF$^{FBXL3}$ and SCF$^{FBXL21}$, regulate ubiquitination and degradation of mCRYs to govern the circadian clock in mammals[4–8]. Similarly, a cullin 1-based E3 ubiquitin ligase receptor, Jetlag, and a cullin 4-based E3 ubiquitin ligase receptor, Brwd3, bind to dCRY to regulate ubiquitination of Tim (Timeless) and dCRY in *Drosophila*[9,10].

*Arabidopsis* cryptochrome 2 (CRY2) is one of the best studied plant CRYs that mediate blue light inhibition of cell elongation and photoperiodic promotion of floral initiation[11,12]. These physiological activities of CRY2 are regulated by at least three blue light-dependent mechanisms. First, CRY2 undergoes blue light-dependent oligomerization to become active tetramers[13–17]. The BIC1 and BIC2 (Blue-light Inhibitors of Cryptochromes 1 and 2) proteins interact with photoexcited CRY2 to negatively regulate CRY2 photooligomerization in light, whereas the active CRY2 homooligomers may also undergo thermal relaxation to become inactive monomers in the absence of light[14–17]. Second, the activity of CRY2 homooligomers are positively regulated by protein phosphorylation reactions catalyzed by four related protein kinases PPKs (Photoregulatory Protein Kinases 1–4)[18–20]. Third, the photoexcited and phosphorylated CRY2 proteins undergo polyubiquitination catalyzed by the E3 ubiquitin ligase Cul4$^{COP1/SPAs}$ and subsequently degraded by the 26S proteasome[11,21,22]. Like animal CRYs, the abundance and overall cellular activity of plant CRYs are regulated by phosphorylation, ubiquitination, and proteolysis[3]. However, only a cullin 4 family E3 ubiquitin ligase, Cul4$^{COP1/SPAs}$, is presently known to regulate ubiquitination and degradation of plant CRYs[3], raising the question how the highly conserved CRYs are differentially regulated in different evolutionary lineages.

Under natural light conditions, plants rely on the coaction of blue light receptors CRYs and the red/far-red light receptors phytochromes phys[23,24] to achieve the optimal photoresponses[25]. Mechanistically, the CRY-phy coaction could be achieved by different photoreceptors physically complexing with the same signaling proteins, such as bHLH transcription factors, Phytochrome Interacting Factors (PIF1-8)[26], Photoregulatory Protein Kinases (PPK1-4)[20,27], the substrate receptors and co-receptors of the Cul4$^{COP1/SPAs}$ E3 ligases, COP1[28] and SPAs[29]. For example, CRY2 interacts with PPKs, which catalyze blue light-dependent phosphorylation of CRY2 to positively regulate the functions, polyubiquitination, and degradation of CRY2[20]. On the other hand, PPKs interact with the phyB-PIF3 complex to negatively regulate the function of phyB by the so-called "mutually assured destruction" mechanism[27,30]. According to this hypothesis, the photoexcited phyB interacts with PIF3, recruiting PPKs to phosphorylate PIF3; phosphorylated PIF3 interacts with the substrate receptors of the Cul3$^{LRBs}$, Light-Response Bric-a-Brack/Tramtrack/Broad (LRB 1-3), which are the substrate receptors of the Cul3$^{LRBs}$ E3 ligases that catalyze polyubiquitination and degradation of PIF3 that also leads to phyB degradation[27]. *Arabidopsis* genome encodes three LRBs, LRB1, LRB2, and LRB3[30,31]. LRB1 and LRB2 share higher homology and higher levels of mRNA expression than LRB3, and they act redundantly to suppress phyB-dependent photoresponses[31]. It remains unclear whether Cul3$^{LRBs}$ play roles in the CRY-dependent signaling process in addition to its important function regulating PIF3 ubiquitination and phyB-dependent red-light responses.

## Results

### LRBs are required for the CRY-dependent blue light responses.
We have previously shown that the blue light-dependent ubiquitination and proteolysis of CRY2 is diminished but not completely abolished in the *cop1* null mutant[22], suggesting the existence of another E3 ubiquitin ligase in addition to Cul4$^{COP1/SPAs}$. We analyzed CRY2 complexomes isolated from transgenic *Arabidopsis* plants overexpressing GFP-CRY2, using the immunoprecipitation-mass spectrometry (IP-MS)[20], and identified LRB1 and LRB2 as CRY2-associated proteins in a blue light-dependent manner (Supplementary Table 1). We analyzed flowering time of the *lrb1lrb2-2lrb3* triple mutant. The *lrb123* triple mutant showed slightly delayed flowering in long day, and it appears somewhat additive to the delay-flowering of the *cry1cry2* mutant (Supplementary Fig. 1). However, given the known role of LRBs as regulators of phytochrome abundance and functions[30,31], and the antagonistic as well as redundant functions of phytochromes and cryptochromes in the control of flowering time[3], it is technically difficult to clarify the exact roles of LRBs in the CRY-dependent control of flowering-time by only the genetic analyses. We then examined whether LRBs play a role in blue light-dependent regulation of seedling development. As reported previously[30], the *lrb123* triple mutant showed no apparent abnormality when grown in darkness or far-red light, but it exhibited a dramatic short-hypocotyl phenotype when grown in continuous red light (Fig. 1a, b). The *lrb123* triple mutant also showed short-hypocotyl phenotype when grown in either long-day or short-day photoperiods illuminated with white light (Fig. 1a, b). These results are consistent with the established function of LRBs regulating red light-dependent activities of phyB[30,31]. When grown in continuous blue light, the *cry1cry2* mutant developed long hypocotyl phenotype as previously reported[32], whereas the *lrb123* showed hypocotyls modestly shorter than that of the wild-type seedings (Fig. 1c, d). The short hypocotyl phenotype of *lrb123* mutant in blue light can be rescued by constitutively overexpressing the LRB2 protein in the *lrb123* mutant (Supplementary Fig. 2). This defective phenotype of the *lrb123* mutant seedlings (6-day-old) is more apparent in seedlings grown in blue light of the light intensities higher than $10 \, \mu mol \, m^{-2} \, s^{-1}$ (Fig. 1d), which is consistent with the previous report that showed hardly distinguished phenotype of the wild-type and the *lrb12* double mutant (4-day-old) grown in continuous blue light with the light intensities of $10 \, \mu mol \, m^{-2} \, s^{-1}$ or lower[31]. These results argue for a light intensity-dependent function of LRBs in blue light. To examine the functional relationship between CRYs and LRBs, we prepared the *cry1cry2lrb123* quintuple mutant and compared the hypocotyl phenotype of the quintuple mutant with its parents (Fig. 1e, f). When grown under continuous blue light, the *cry1cry2* mutant and the *lrb123* mutant exhibit long or short hypocotyl phenotype, respectively. However, the *cry1cry2lrb123* quintuple mutant shows the long hypocotyl phenotype indistinguishable from that of the *cry1cry2* mutant parent in blue light (Fig. 1e, f), but not in other light conditions (Fig. 1a, b). This result demonstrates that the function of LRBs in blue light is dependent on the activity of the blue-light receptor CRYs. Comparing with the *lrb123* parent, the *lrb123cop1* quadruple mutant shows de-etiolated phenotype of the *cop1* parent in darkness but a slight additive effect of the *lrb123* and *cop1* mutants in blue light (Fig. 1g, h). Taken together, these results argue that LRBs regulate plant development by controlling not only the red light-dependent activity of phytochromes but also the blue light-dependent activity of cryptochromes.

### LRBs are required for the blue light-dependent CRY2 degradation.
We next tested how LRBs regulate cryptochrome activity by examining the blue light-dependent degradation of CRY2 in

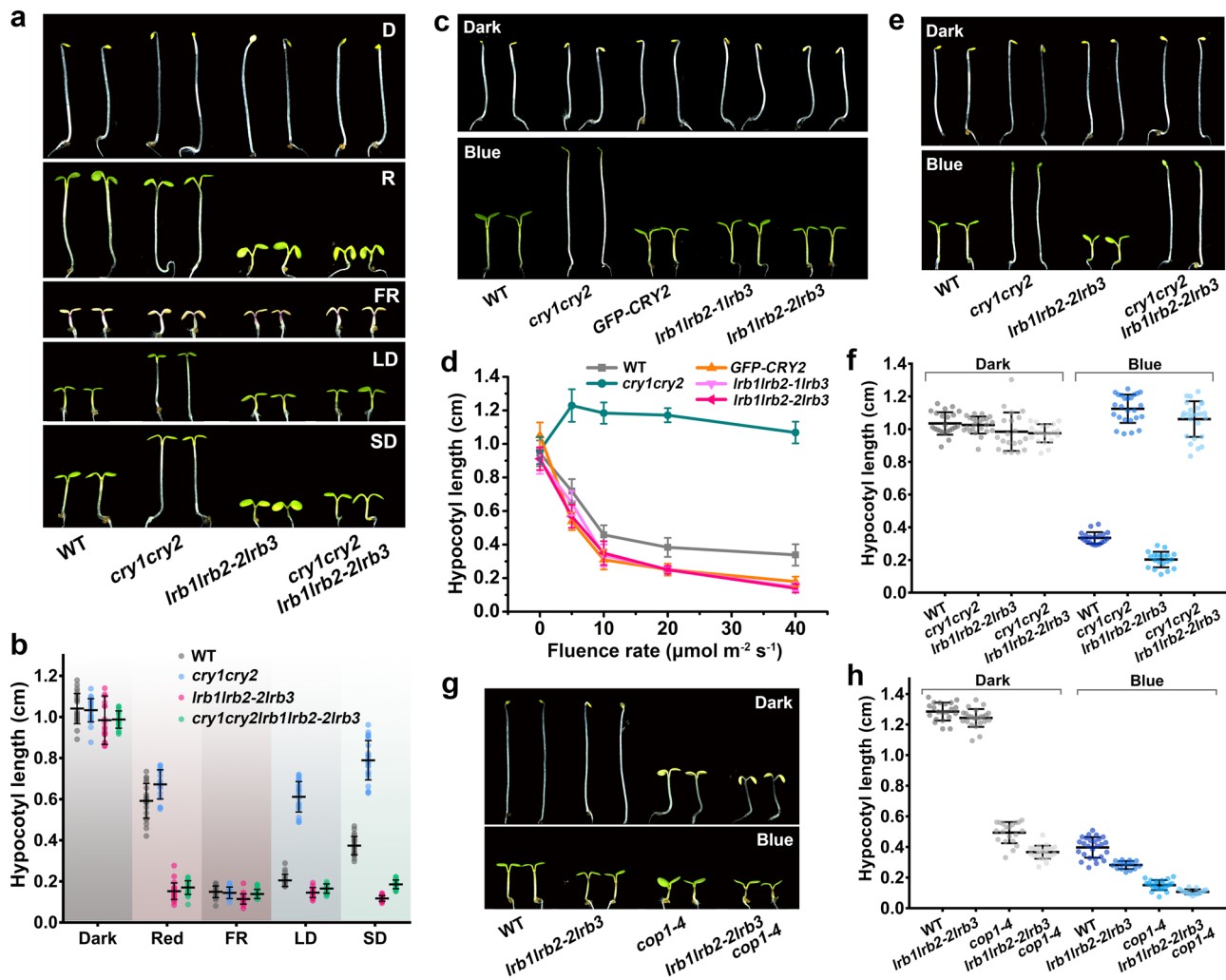

**Fig. 1 LRBs are required for blue light responses. a** 6-day-old seedlings grown in darkness (D), red light (R, 20 μmol m⁻² s⁻¹), far-red light (FR, 6 μmol m⁻² s⁻¹), long days (LD, 16 h light / 8 h dark), or short days (SD, 8 h light / 16 h dark). **b** Measurements of hypocotyl length of seedlings shown in (**a**), (mean ± SD). **c** 6-day-old seedlings grown in darkness or blue light (10 μmol m⁻² s⁻¹). **d** Measurements of hypocotyl length of indicated genotypes grown under blue light of different intensities (0, 5, 10, 20, and 40 μmol m⁻² s⁻¹) for 6 days, (mean ± SD, $n = 25$). **e, g** 6-day old seedlings grown in darkness or blue light (20 μmol m⁻² s⁻¹). **f, h** Measurements of hypocotyl length of seedlings shown in (**e**) and (**g**), (mean ± SD). The above experiments were repeated at least three times with similar results.

the *lrb123* mutants. Results shown in Fig. 2 demonstrate that LRBs are required for the blue light-induced degradation of CRY2. The CRY2 protein level decreased by at least 70% (in 30 min) to 85% (in 60 min) in etiolated wild-type seedlings transferred to blue light (30 μmol m⁻² s⁻¹) (Fig. 2a, b). In contrast, levels of the CRY2 protein exhibited little change in one *lrb123* triple mutant allele (*lrb1lrb2-1lrb3*) or <50% decrease in the second *lrb123* triple mutant allele (*lrb1lrb2-2lrb3*) under the same condition (Fig. 2a, b). This result indicates that *lrb1lrb2-1lrb3* is a slightly stronger allele in comparison to *lrb1lrb2-2lrb3*. We next compared the blue light-induced CRY2 degradation in *lrb123*, *cop1*, and *lrb123cop1* quadruple mutant in seedlings grown in continuous dark or blue light. CRY2 shows similar abundance in the wild-type and both *lrb123* triple mutant alleles grown in continuous blue light, whereas the level of CRY2 is higher than that of the wild-type in the *cop1* mutant grown in continuous blue light (Fig. 2c, d). Results of this experiment indicate that COP1, but not LRBs, determines abundance of the CRY2 protein in seedlings exposed to prolonged blue light. We hypothesize that LRBs are responsible for the rapid blue light response of CRY2 proteolysis but not the prolonged photoresponse of

CRY2 proteolysis, whereas COP1 is required for the prolonged photoresponse of CRY2 proteolysis. This difference may be explained, at least partially, by the respective roles of Cul3^LRBs and Cul4^COP1/SPAs ligases in CRY2 signal transduction. COP1 is a major signaling molecule of CRY2[33–35], in addition to its role as a E3 ligase of CRY2[18], but LRBs are not known to play a direct role in the CRY2 signaling transduction. It is conceivable that the CRY2 suppression of COP1 activity in the early stage of light exposure, including COP1-dependent CRY2 ubiquitination, may be partially overcome by some feedback regulations during prolonged light exposure. We further tested the hypothesis that LRBs and COP1 may regulate CRY2 degradation under different light conditions, by comparing the photoresponsive CRY2 proteolysis in etiolated seedlings transferred to blue light from 30 min to 14 h (Fig. 2e, f). In etiolated seedlings exposed to blue light, CRY2 degradation shows clear defect in the *lrb123* triple mutants within 2 h of light exposure, but the *cop1* mutant exhibits more pronounced defect in CRY2 degradation during prolonged light exposure. These results are consistent with the observations that more CRY2 accumulated in the *cop1* mutant in continuous blue light, but not in the *lrb123* mutant alleles (Fig. 2c, d). Importantly,

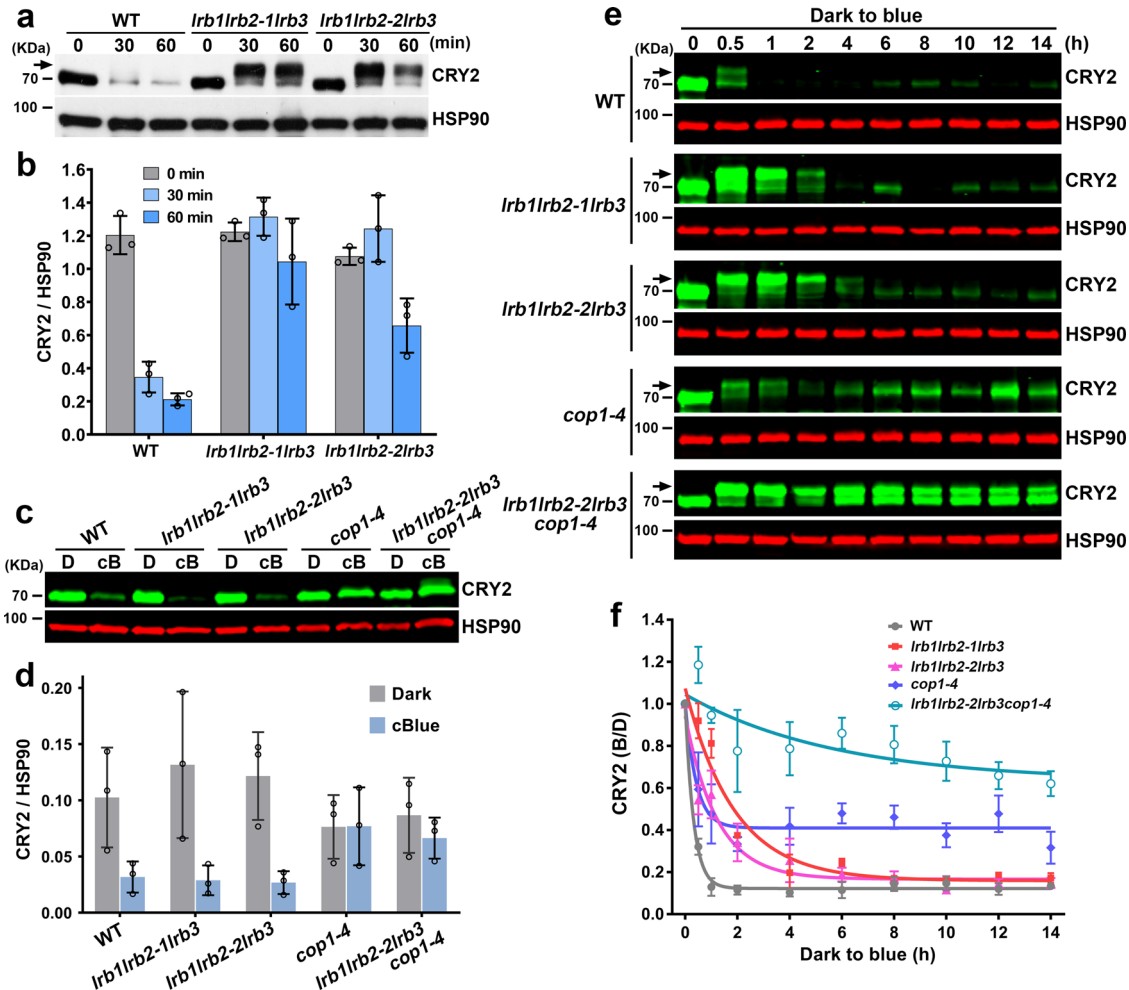

**Fig. 2 LRBs and COP1 regulate the rapid or prolonged proteolysis of CRY2, respectively. a** Representative immunoblots showing the abundance of endogenous CRY2 in the 7-day-old etiolated wild type (WT) and *lrb123* mutants irradiated with 30 μmol m$^{-2}$ s$^{-1}$ of blue light for the indicated time. **b** Quantitative analysis of CRY2 abundance from immunoblots shown in (**a**). Data are presented as mean ± SD (*n* = 3 individual immunoblots). **c** Representative immunoblots showing the abundance of endogenous CRY2 in seedlings of indicated genotypes grown in darkness (D) or continuous blue light (cB, 30 μmol m$^{-2}$ s$^{-1}$). **d** Quantitative analysis of CRY2 abundance from immunoblots shown in (**c**). Data are presented as mean ± SD (*n* = 3 individual immunoblots). **e** Representative immunoblots showing degradation of the endogenous CRY2 in 7-day-old etiolated seedlings irradiated with 30 μmol m$^{-2}$ s$^{-1}$ of blue light for indicated time. **f** Quantitative analysis of CRY2 degradation from immunoblots shown in (**e**). Data are presented as mean ± SD (*n* = 3 individual immunoblots). CRY2 (B/D) = [CRY2/HSP90]$^{blue}$ / [CRY2/HSP90]$^{dark}$. The best fitted curves with one phase decay of nonlinear regression were shown. CRY2 and HSP90 were detected with anti-CRY2 antibody and anti-HSP90 antibody, respectively. HSP90 is used as a loading control. Arrows indicate phosphorylated CRY2. The above experiments were repeated at least three times with similar results.

little CRY2 degradation was detected in the *lrb123cop1* quadruple mutant for both short or long exposure of blue light (Fig. 2c–f). We concluded that LRBs and COP1 are both required for the blue light-induced CRY2 degradation.

**LRBs interact with photoexcited and phosphorylated CRY2.** Given that a substrate receptor of E3 ubiquitin ligases must physically interact with the substrate, we examined the CRY2/LRB interaction (Fig. 3), using the co-immunoprecipitation (co-IP) assays of proteins co-expressed in the heterologous HEK293T (Human Embryo Kidney 293T) cells as we previously reported[14]. Because we have previously shown that the blue light-induced phosphorylation of CRY2 is required for CRY2 ubiquitination and degradation, and that PPK1 is one of the four related protein kinases that specifically phosphorylate photoexcited CRY2[18,20], we tested CRY2/LRB interaction in the presence of the wild-type PPK1 in response to blue light in HEK293T cells. Results of this experiment show that LRB1 and LRB2 preferentially interact with

phosphorylated CRY2 in HEK293T cells co-expressing PPK1, but only when cells were exposed to blue light (Fig. 3a, b). LRB2 failed to interact with the photo-insensitive CRY2$^{D387A}$ mutant that does not bind the FAD (Flavin Adenine Dinucleotide) chromophore[36] (Fig. 3c), which is consistent with the light dependence of CRY2/LRB interaction. LRB2 failed to interact with CRY2 in light-treated HEK293T cells co-expressing the catalytically inactive PPK1$^{D267N}$ (Fig. 3d), which is consistent with the phosphorylation dependence of CRY2/LRB interaction. These results support a hypothesis that LRBs directly and specifically interact with photoexcited and phosphorylated CRY2.

We next examined the CRY2/LRB interaction in plant cells, using the BiFC (Bimolecular Fluorescence Complementation) assay in *Arabidopsis* leaves[37] (Fig. 4a) and split-luciferase assay in tobacco (*Nicotiana benthamiana*) leaves (Fig. 4b, c). Results of the BiFC assay show strong BiFC signals between CRY2 and LRB1 or CRY2 and LRB2 (Fig. 4a). In contrast, no BiFC signal was detected between the photo-insensitive CRY2$^{D387A}$ and LRB1 or CRY2$^{D387A}$ and LRB2 (Fig. 4a). These results clearly demonstrate

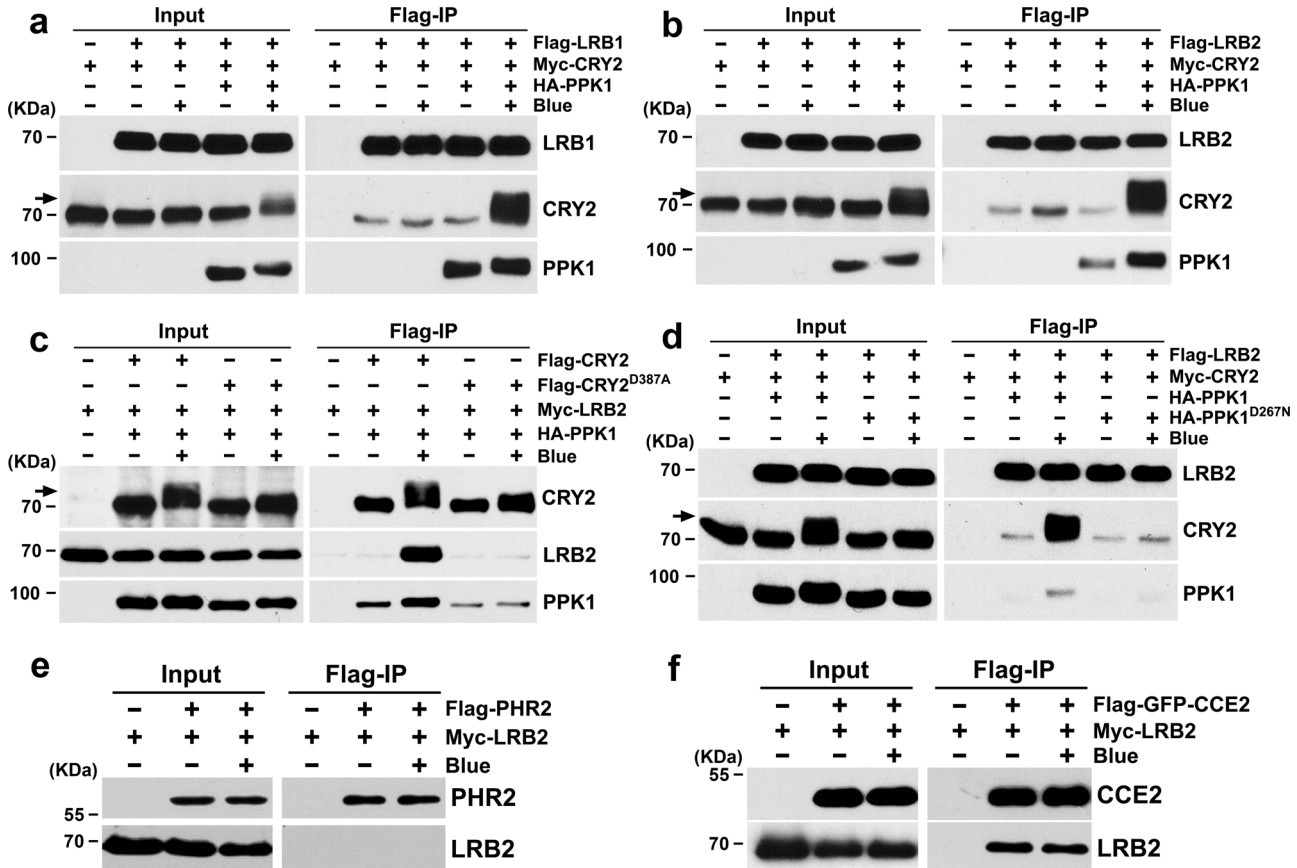

**Fig. 3 LRBs interact with phosphorylated CRY2 in the HEK293T cells. a** Co-immunoprecipitation (co-IP) assays showing the blue light and phosphorylation-dependent interaction of LRB1 and CRY2 in heterologous HEK293T cells. Immunoprecipitations (IP) were performed with Flag-conjugated beads. The IP (LRB1) and co-IP (CRY2) products were detected with anti-Flag and anti-CRY2 antibodies, respectively. PPK1 was detected with anti-HA antibody. **b** Co-IP assays showing the blue light and phosphorylation-dependent interaction of LRB2 and CRY2 in HEK293T cells. The experiments were performed as in (**a**). **c** Co-IP assays showing that LRB2 failed to interact with the photo-insensitive CRY2$^{D387A}$ mutant. The experiments were performed as in (**a**), except that the IP (CRY2 and CRY2$^{D387A}$) and co-IP (LRB2) products were detected with anti-CRY2 and anti-Myc antibodies, respectively. **d** Co-IP assays showing the phosphorylation-dependent interaction of LRB2 and CRY2 interaction in HEK293T cells. The experiments were performed as in (**a**). PPK1$^{D267N}$, catalytically inactive PPK1. **e–f** Co-IP assays showing the interaction between CRY2 CCE domain and LRB2. IP was performed with Flag-conjugated beads. The IP (PHR2 and CCE2) and co-IP (LRB2) products were detected with anti-Flag and anti-Myc antibodies, respectively. PHR2, photolyase homologous region of CRY2; CCE2, CRY2 C-terminal extension. The cells were treated with blue light (+ Blue; 100 µmol m$^{-2}$ s$^{-1}$) for 2 h or kept in the dark (-Blue). Arrows show phosphorylated CRY2. The above experiments were repeated three times with similar results.

the direct interaction of CRY2 and LRB proteins in vivo. Consistently, the split-luciferase assays suggest the direct interaction between CRY2 and LRBs that contains deletion of the BTB domain (Fig. 4b, c). Given the common affinity of BTB to Cullin 3 (Cul3)[31,38], this experiment argues that the CRY2/LRB interaction is independent from BTB domain of LRBs or LRB/Cullin interaction. CRY2/LRB complexes are also detected by the co-IP (co-immunoprecipitation) assays in transgenic *Arabidopsis* seedlings stably co-expressing Flag-GFP-tagged CRY2 (FGFP-CRY2) and Myc-LRB1 or FGFP-CRY2 and Myc-LRB2 (Fig. 4d, e). The co-IP assays detected increasing amount of the CRY2/LRB complexes in etiolated seedlings exposed to blue light or in seedlings grown in LD and continuous white light (Fig. 4d, e). These results support a hypothesis that LRBs complex with CRY2 in a light-dependent manner and that LRBs act as the substrate receptors of Cul3$^{LRBs}$ for the blue light-dependent CRY2 ubiquitination. It is noticed that the extent of photo-promotion of the accumulation of CRY2/LRB complexes in *Arabidopsis* seedlings is not as strong as that detected in light-treated heterologous HEK293T cells in the presence of PPK1 (Fig. 3) or that of blue light-induced CRY2 homooligomerization detected in *Arabidopsis* (Fig. 4d, e). These results are consistent with the

notions that CRY2 oligomerization is the primary photoreaction of the CRY2 photoreceptor and that LRBs interact with CRY2 to facilitate CRY2 degradation in vivo, but not in heterologous HEK293T cells.

**LRBs and COP1 are required for the photoresponsive polyubiquitination of CRY2.** To test whether the Cul3$^{LRBs}$ and Cul4$^{COP1/SPAs}$ E3 ubiquitin ligases catalyze blue light-dependent CRY2 ubiquitination, we performed three independent experiments. In the first two experiments, we prepared transgenic plants overexpressing FGFP-CRY2 in the wild-type (WT), the strong (*lrb1lrb2-1lrb3*) or weak (*lrb1lrb2-2lrb3*) alleles of the *lrb123* triple mutant (Fig. 5a, b, d), and the *cop1* mutant backgrounds (Fig. 5c, d). In the third experiment, we prepared double-tagged transgenic lines that overexpress FGFP-CRY2 with either the Myc-tagged LRB (Myc-LRB1, Myc-LRB2) or the Myc-tagged COP1 (Myc-COP1) (Fig. 5e). In the first experiment, polyubiquitinated proteins were indiscriminately precipitated using the TUBE2 (Tandem Ubiquitin Binding Entity 2)-conjugated beads[39], and the TUBE2-enriched proteins were analyzed by immunoblots probed with the anti-ubiquitin or anti-CRY2 antibodies (Fig. 5a–c). This experiment detected a high level of polyubiquitination of CRY2 in

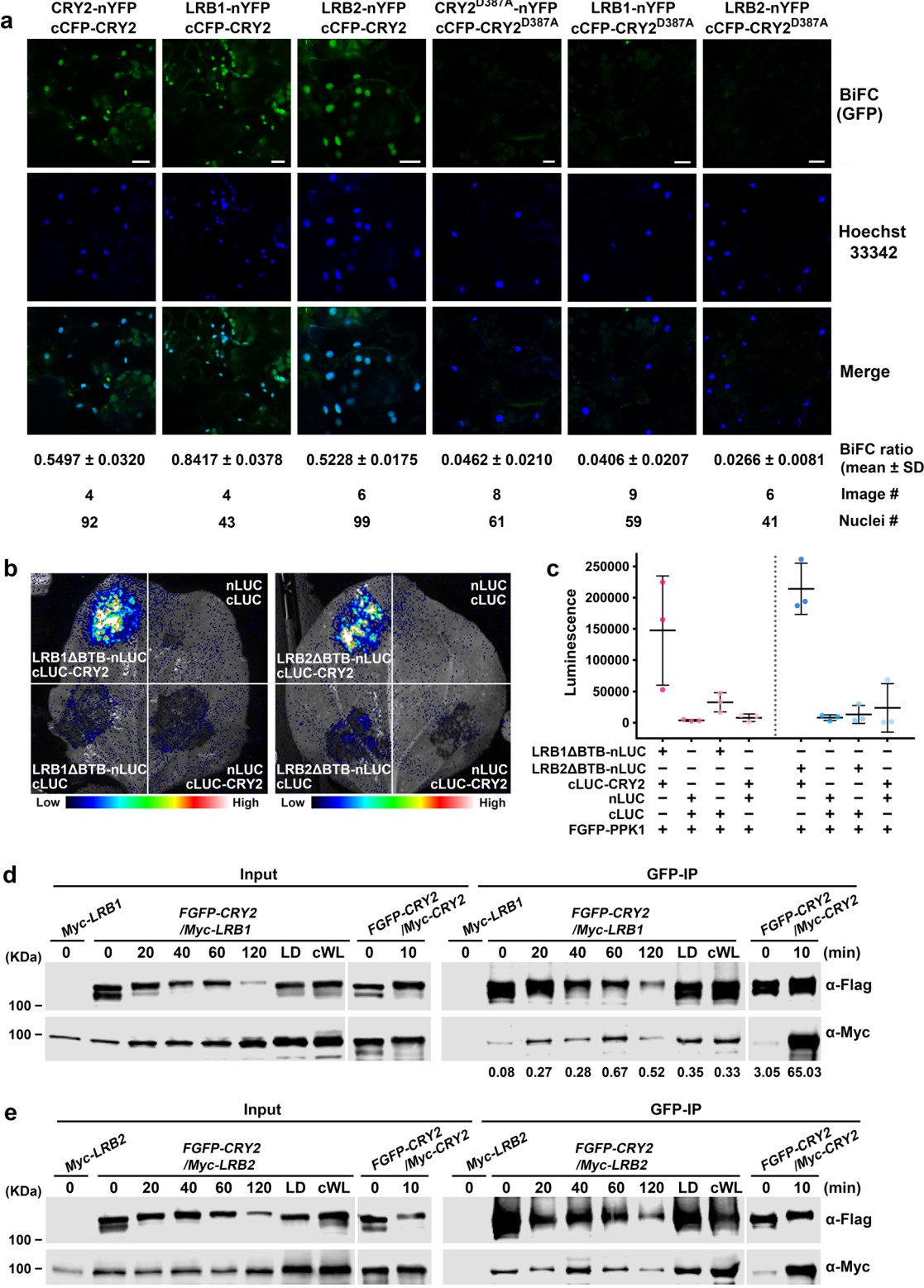

blue light-treated wild-type background seedlings (Fig. 5a–c). However, little ubiquitinated CRY2 was detected in etiolated seedlings or in blue light-treated *lrb123* or *cop1* mutant background seedlings (Fig. 5a–c). In the other two experiments, FGFP-CRY2 were immunoprecipitated by the GFP-trap beads, and the level of polyubiquitination of FGFP-CRY2 was analyzed by immunoblots probed against anti-ubiquitin antibody (Fig. 5d, e). In etiolated seedlings exposed to blue light, a high level of

ubiquitinated FGFP-CRY2 was detected in the wild-type seedlings overexpressing FGFP-CRY2. In contrast, a lower level of ubiquitinated FGFP-CRY2 was detected in *lrb1lrb2-2lrb3* weak allele, whereas almost no ubiquitinated FGFP-CRY2 was detected in the *lrb1lrb2-1lrb3* strong allele or the *cop1* mutant seedlings transgenically overexpressing FGFP-CRY2 (Fig. 5d). In comparison to FGFP-CRY2 overexpressed in the wild-type background, the blue light-induced polyubiquitination of FGFP-CRY2 is enhanced in

**Fig. 4 LRBs interact with CRY2 in vivo. a** Confocal microscopic images showing BiFC signals of indicated protein pairs transiently expressed in *Arabidopsis* leaves. The Hoechst 33342 (which is used to stain the nuclei) and GFP (BiFC) signals are shown. The relative intensity of CRY2/LRB interaction was presented as BiFC ratio, calculated as BiFC ratio = [GFP intensity]$^{nuclei}$ / [Hoechst 33342 intensity]$^{nuclei}$. BiFC ratio (mean ± SD), total number of quantified images and nuclei are shown blow the images. CRY2$^{D387A}$, photo-insensitive CRY2, is used as the negative control. Scale bars, 10 μm. **b** Split-luciferase complementation assays showing the interactions of LRB1 and CRY2 or LRB2 and CRY2 in tobacco. Indicated split-LUC protein pairs were transiently co-expressed in *N. benthamiana* with FGFP-PPK1. LRB1ΔBTB, LRB1 with deletion of BTB domain (143-212 aa); LRB2ΔBTB, LRB2 with deletion of BTB domain (145-250 aa). **c** Quantification of split-luciferase complementation assays shown in (**b**). Data was shown as mean ± SD (*n* = 3 individual experiments). **d**–**e** Co-immunoprecipitation assays showing the interactions of LRB1 and CRY2 (**d**) or LRB2 and CRY2 (**e**) in *Arabidopsis*. 7-day-old seedlings expressing Myc-LRB or co-expressing FGFP-CRY2 and Myc-LRB were grown in the dark and then treated with blue light (100 μmol m$^{-2}$ s$^{-1}$) for the indicated time, or grown in long days (LD, 16 h light/ 8 h dark) and continuous white light (cWL). Plant extracts were immunoprecipitated by GFP-trap beads. The IP and co-IP products were detected with anti-Flag and anti-Myc antibodies, respectively. Co-IP experiments of 7-day-old seedlings co-expressing FGFP-CRY2 and Myc-CRY2 were performed in parallel as positive controls. The relative interaction intensity of CRY2/LRB or CRY2/CRY2 is calculated by [Co-IP intensity] / [IP intensity] and shown blow the immunoblots. The above experiments were repeated twice with similar results.

plants co-overexpressing LRB1, LRB2, or COP1 (Fig. 5e). Longer exposures of the chemiluminescence (Fig. 5d, bottom) demonstrate residual polyubiquitinations of CRY2 in either *lrb123* or *cop1* mutant and partial functional redundancy of the two E3 ligases. The minor ubiquitination signals detected in etiolated seedlings may result from basal activities of the enzymes or background signals. Taken together, results of those three independent experiments all support the hypothesis that two distinct E3 ubiquitin ligases, Cul3$^{LRBs}$ and Cul4$^{COP1/SPAs}$, can both catalyze blue light-dependent polyubiquitination of CRY2 in vivo.

**COP1 and LRBs interact with different structural elements of CRY2.** We wish to understand different mechanisms governing the activities of COP1 and LRBs with respect to CRY2 ubiquitination. Most presently known light-dependent CRY2-interacting proteins interact with the photon-sensing PHR domain of CRY2[3]. However, COP1 is one of the few exceptions. COP1 binds to the CCE domain of CRYs regardless of light in vitro or in yeast cells[33–35], although COP1 binds to CRY2 in the light-dependent manner in vivo[40,41]. Like COP1, LRB2 also interacts with the CCE domain of CRY2 in heterologous HEK293T cells (Fig. 3e, f). However, in contrast to COP1 that interacts with CRYs regardless of light in vitro or in yeast cells, LRBs preferentially interact with phosphorylated CRY2 in the blue light-dependent manner in HEK293T cells (Fig. 3a–d). Given that analyses of molecular interactions of plant proteins in heterologous yeast cells, HEK293T cells, or in vitro are essentially equivalent, we reasoned that CRY2 may interact with COP1 or LRBs via different structural elements and/or respond to the photoresponsive CRY2 conformation changes differently.

We investigated this question by analyses of the CRY2$^{P532L}$ mutation that is impaired at the VP motif (Fig. 6a). The VP motif is a conserved COP1-binding motif of many substrates of the Cul4$^{COP1/SPAs}$ E3 ligase, including CRY2[42,43]. We previously reported that the CRY2$^{P532L}$ mutation impaired at the VP motif (Fig. 6a) loses all the physiological activities tested but retains many photochemical properties of CRY2, including blue light-induced oligomerization, phosphorylation, ubiquitination, and degradation[15]. We first compared the relative activity of FGFP-CRY2 and FGFP-CRY2$^{P532L}$, using the hypocotyl inhibition assay in transgenic plants expressing the respective recombinant protein at similar levels in the *cry1cry2* mutant background (Supplementary Fig. 3a). Transgenic seedlings expressing FGFP-CRY2$^{P532L}$ showed no obvious activity mediating light inhibition of hypocotyl elongation under all light intensities of blue light tested (Fig. 6b, Supplementary Fig. 3b). This result confirmed our previous report that *CRY2$^{P532L}$* is a loss-of-function mutation[15]. We then compared the kinetics of blue light-induced degradation of FGFP-CRY2 and FGFP-CRY2$^{P532L}$, using the same transgenic

lines (Fig. 6c, d). In transgenic seedlings, the mutant recombinant protein FGFP-CRY2$^{P532L}$ is degraded in response to blue light as we previously reported[15], although it appears to degrade slower than its wild-type counterpart (Fig. 6c, d). The apparent half-life of FGFP-CRY2$^{P532L}$ is more than twice that of FGFP-CRY2 (Fig. 6d). When co-expressed with PPK1 in HEK293T cells, both the wild-type CRY2 and the CRY2$^{P532L}$ mutant protein exhibits clear blue light-dependent phosphorylation (Fig. 6e, f). In HEK293T cells co-expressing SPA1, COP1 interacts with CRY2 in the light-dependent manner with both phosphorylated and unphosphorylated CRY2, whereas neither the photo-insensitive CRY2$^{D387A}$ mutant nor the COP1-insensitive CRY2$^{P532L}$ mutant interact with COP1 or SPA1 (Fig. 6e). LRB2 preferentially interacts with photoexcited and phosphorylated CRY2, as well as the photo-excited and phosphorylated CRY2$^{P532L}$ mutant to the similar extent (Fig. 6f). These results are consistent with that CRY2$^{P532L}$ remains photoresponsive in vivo[15] and that CRY2$^{P532L}$ undergoes blue light-dependent ubiquitination and degradation in vivo (Fig. 6c, d). Since the CRY2$^{P532L}$ mutant does not interact with COP1, these results confirm that COP1 and LRB2 interact with different structural elements of CRY2. Taken together, results of our analyses of the CRY2$^{P532L}$ mutant further supports the hypothesis that Cul3$^{LRBs}$ is the second type of E3 ligase required for blue light-dependent CRY2 ubiquitination and degradation.

The notion that Cul3$^{LRBs}$ and Cul4$^{COP1/SPAs}$ regulate CRY2 predominantly in the early or prolonged state of light exposure, respectively, seems inconsistent with the kinetics of CRY2$^{P532L}$ degradation in vivo (Fig. 6c, d). Because the CRY2$^{P532L}$ interacts with LRB2 but not COP1 (Fig. 6e, f), its degradation is expected to mimic that of the wild-type CRY2 in the *cop1* mutant (Fig. 2). Instead, the kinetics of CRY2$^{P532L}$ degradation is more like that of the wild-type CRY2 in the *lrb123* triple mutant (Fig. 2). This seemingly perplexing phenomenon may be explained by the unusual photochemical characteristics of the CRY2$^{P532L}$ mutant. Photoexcited CRY2$^{P532L}$ monomers exhibit increased affinity among themselves in comparison to the wild-type CRY2 (Fig. 6g, h) without obvious change of the rate or affinity in the dark-reversion reaction (Fig. 6i, j). Because photooligomerization is necessary for all light-dependent reactions of CRY2, including photoresponsive ubiquitination and degradation[14,15], it is conceivable that the unusual photochemical characteristics of the CRY2$^{P532L}$ mutant may change its relationship with various regulators in vivo, including LRBs or 26S proteosomes, to alter its degradation kinetics in the unexpected manner. Taken together, the results of our experiments argue that two types of substrate receptors, COP1 and LRBs, interact with CRY2 via distinct structural elements of CRY2 to enable the blue light-induced CRY2 ubiquitination and degradation by the respective E3 ubiquitin ligases, Cul3$^{LRBs}$ and Cul4$^{COP1/SPAs}$.

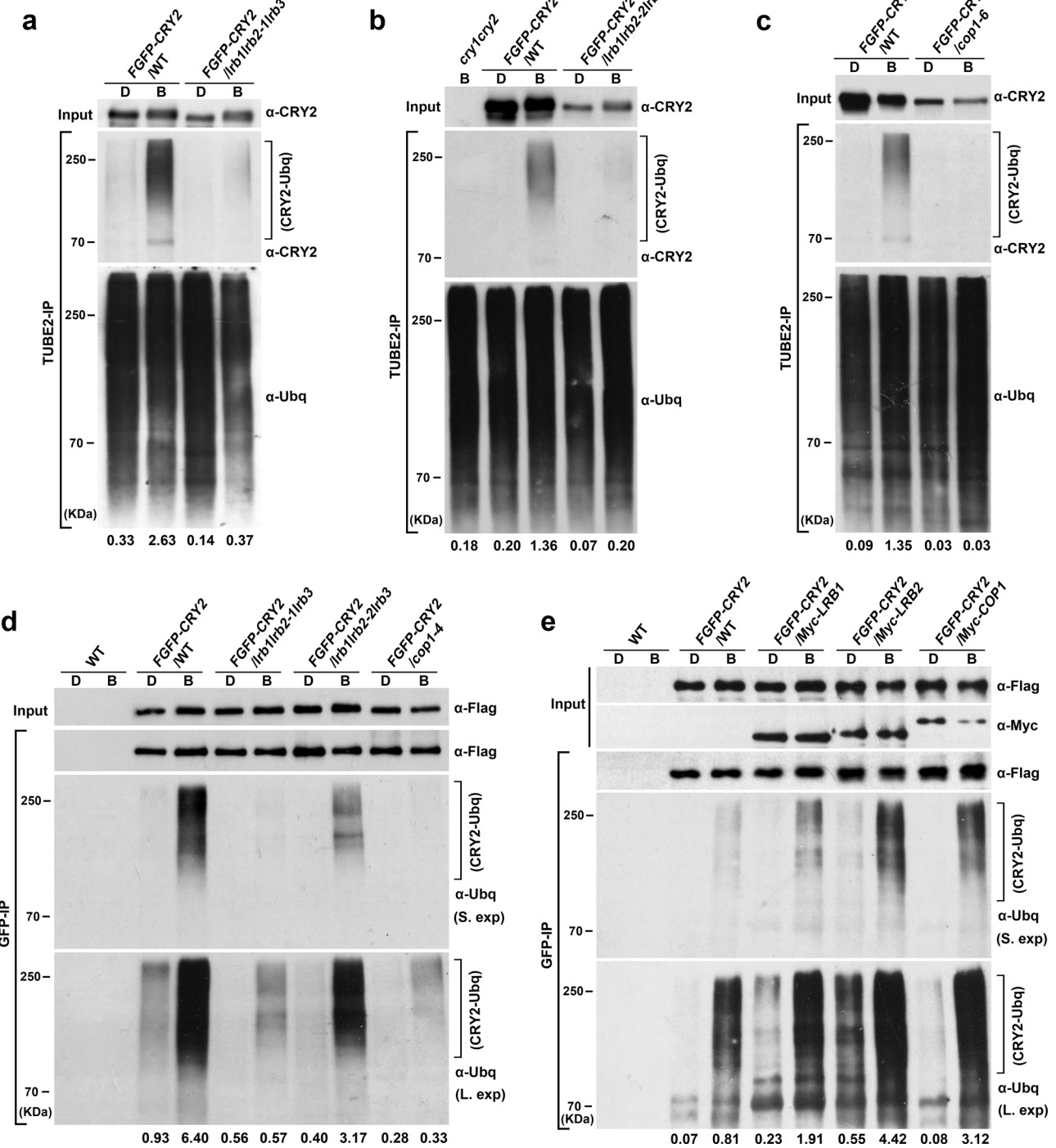

**Fig. 5 LRBs and COP1 are both required for CRY2 ubiquitination.** Immunoblots showing the ubiquitination of FGFP-CRY2 in indicated genotypes. 7-day-old etiolated seedlings constitutively expressing FGFP-CRY2 in indicated genotypes were pretreated with MG132 and kept in the dark or exposed to 30 μmol m$^{-2}$ s$^{-1}$ of blue light for 5, 10, and 15 min. The blue light treated samples (5, 10, and 15 min) were pooled together while performing immunoprecipitation for (**a–e**). **a–c** Total ubiquitinated proteins were purified by TUBE2-conjugated beads. Immunoprecipitated proteins were analyzed by immunoblots probed with anti-ubiquitin antibody (α-Ubq) or anti-CRY2 antibody (α-CRY2). The extent of CRY2 ubiquitination was shown below the immunoblots, calculated as [CRY2-Ubq intensity]$^{IP}$/[Ubq intensity]$^{IP}$. **d–e** FGFP-CRY2 proteins were purified with GFP-trap beads. Immunoprecipitated proteins were analyzed by immunoblots probed with anti-ubiquitin antibody (α-Ubq), anti-Flag antibody (α-Flag) or anti-Myc antibody (α-Myc) for detecting epitope-tagged proteins. The extent of CRY2 ubiquitination is calculated as [CRY2-Ubq intensity]$^{IP}$/[Flag intensity]$^{IP}$ with the short exposure immunoblots and shown below the immunoblots. S. exp or L. exp: short or long chemiluminescence exposures of immunoblots. D, dark treatment; B, blue light treatment. The above experiments were repeated at least twice with similar results.

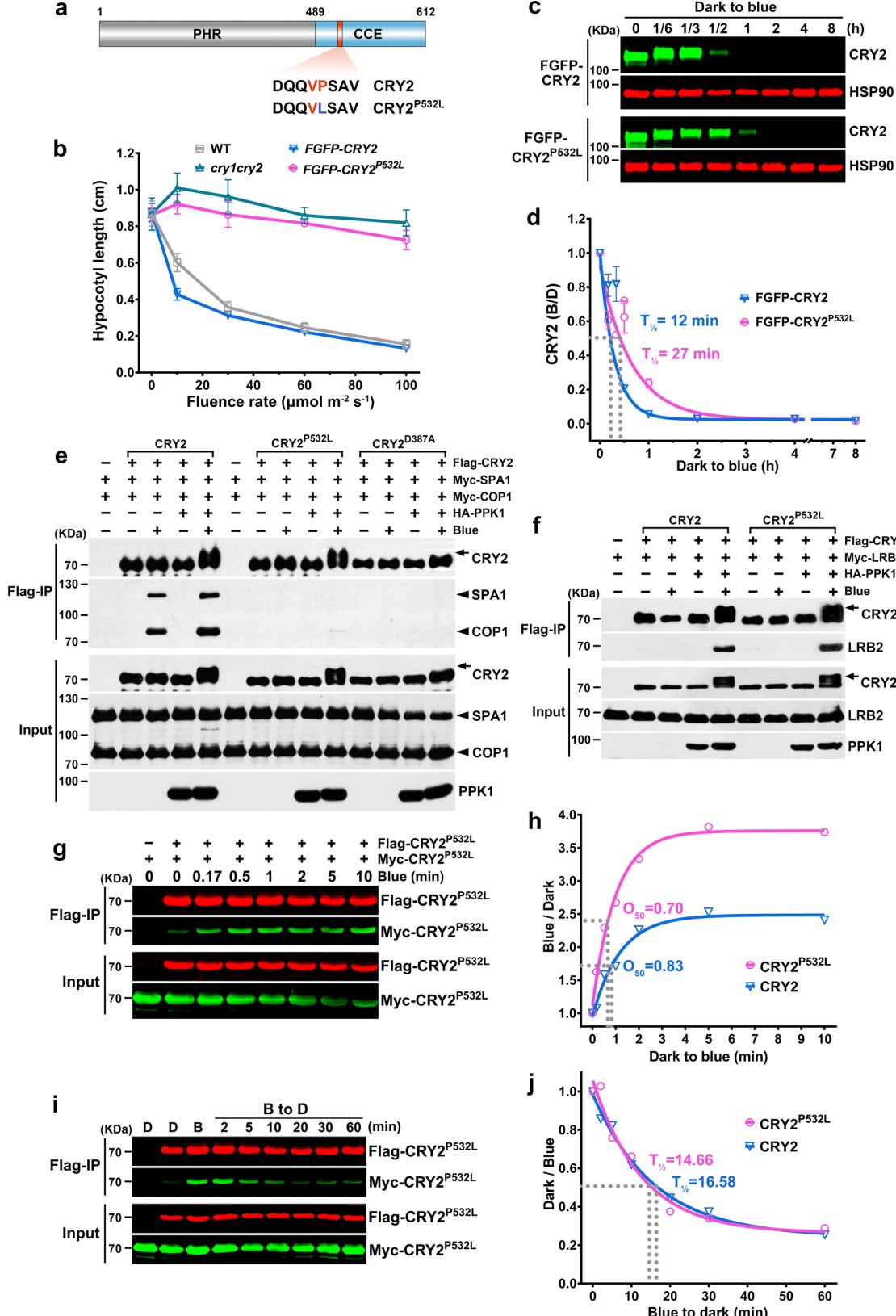

## Discussion

In this study, we demonstrate that, in addition to Cul4$^{COP1/SPAs}$, the E3 ubiquitin ligases Cul3$^{LRBs}$ is required for blue light regulation and function of CRY2 (Fig. 7a). Cul3$^{LRBs}$ appears capable of targeting photoexcited and phosphorylated CRY2 for a rapid ubiquitination and degradation, whereas Cul4$^{COP1/SPAs}$ is responsible for the sustained ubiquitination and degradation of CRY2 in plants under prolonged light exposure. The differential regulation of CRY2 turnover in response to different light conditions appears necessary for the optimal photoresponses of plants grown in nature. LRBs and COP1 interact with different structural elements of CRY2, which explain, at least partially, the different kinetic features of the

**Fig. 6 LRBs and COP1 interact with different structural elements of CRY2. a** A schematic diagram depicting the CRY2[P532L] mutation. PHR, photolyase homologous region; CCE, CRY C-terminal extension; DQQVPSAV, VP motif in CRY2; numbers, amino acid positions. **b** Hypocotyl length of 6-day-old seedlings of indicated genotypes grown under blue light of different light intensities. Data were shown as mean ± SD. **c** Immunoblots showing degradation of FGFP-CRY2 or FGFP-CRY2[P532L] in 7-day-old etiolated seedlings irradiated with blue light (100 μmol m$^{-2}$ s$^{-1}$) for the indicated time. Immunoblots were probed with anti-CRY2 and anti-HSP90 antibodies. **d** Quantitative analysis of CRY2 degradation from immunoblots shown in (**c**), $n = 3$ individual immunoblots. The best fitted curves with one phase decay of nonlinear regression were shown. CRY2 (B/D) = [CRY2/HSP90]$^{blue}$ / [CRY2/HSP90]$^{dark}$. $T_{1/2}$ indicates the time required for 50% degradation. **e–f** Co-IP assays showing the lack of CRY2[P532L]/COP1 interaction (**e**) and the blue light- and phosphorylation-dependent CRY2[P532L]/LRB2 interaction (**f**) in HEK293T cells. Transfected cells were either kept in the dark (− Blue) or treated with 100 μmol m$^{-2}$ s$^{-1}$ blue light for 2 h (+ Blue). CRY2[D387A] is used as a negative control. Anti-Flag, anti-Myc or anti-HA antibodies were used for detecting indicated tagged proteins. Arrows indicate phosphorylated CRY2. **g** Co-IP results showing the photooligomerization of CRY2[P532L] in HEK293T cells. Transfected cells were irradiated with blue light (30 μmol m$^{-2}$ s$^{-1}$) for indicated time. CRY2[P532L] tagged with Flag or Myc were detected with anti-Flag and anti-Myc antibodies, respectively. **h** Comparison of photooligomerization kinetics of CRY2 and CRY2[P532L] in HEK293T cells. Co-IP (α-Myc) signals was normalized to the corresponding IP (α-Flag) signals, and the dark oligomerization level was set as 1. $O_{50}$ indicates the time required to reach 50% saturation of CRY2 photooligomerization. The best fitted curves with one phase association of nonlinear regression were shown. **i** Co-IP results showing the dark reversion of CRY2[P532L] photooligomers in HEK293T cells. The cells were treated with 30 μmol m$^{-2}$s$^{-1}$ blue light for 5 min then moved to the darkness for indicated time. **j** Comparison of the dark reversion dynamics of CRY2 and CRY2[P532L] photooligomers in HEK293T cells. Similar analyses were performed as in (**h**), except that the oligomerization level in blue was set as 1. $T_{1/2}$ indicates the time required to reverse 50% of CRY2 photooligomers into monomers. The best fitted curves analyzed with one phase decay of nonlinear regression were shown. CRY2 photooligomerization and dark reversion data shown in (**h**) and (**j**) were extracted from the published paper[15] with permission. The above experiments were repeated at least twice with similar results.

LRB- and COP1-dependent CRY2 degradation in response to blue light. Photoresponsive expression of LRBs may also play a role in the light regulation of CRY2 turnover. Blue light positively regulates LRB1 and LRB2 protein expression (Fig. 7b–e). The abundance of recombinant LRB proteins in transgenic plants expressing LRB1 or LRB2 driven by either the respective native promoters or the ACT2 constitutive promoter increase in response to blue light. Because Cul4$^{COP1/SPAs}$-dependent ubiquitination and 26S proteosome pathway is best known for degradation of photo-signaling proteins in darkness, it is tempting to speculate that Cul4$^{COP1/SPAs}$ may play a role in the regulation of the LRB proteins expression, and the CRY2-mediated light inhibition of Cul4$^{COP1/SPAs}$ may lead to increased LRB protein accumulation, accelerated turnover of not only photoexcited CRY2 but also photoexcited phyB, and sustained overall photosensitivity of plants grown in nature (Fig. 7a). However, this hypothesis remains to be tested directly.

LRBs and COP1 apparently interact with CRY2 differently: LRBs interacts with photoexcited and phosphorylated CRY2, whereas COP1 interacts with CRY2 regardless of its phosphorylation states. This result would be consistent with CRY2 interacting with COP1 or LRBs via different structural elements. Photoexcitation of CRYs may result in different conformational changes and interactions with different proteins. First, photoexcited CRYs may undergo initial conformational change associated with CRY oligomerization, which have been directly shown by several recent structural studies of the PHR domain of CRY2[3,16,17,44]. Second, light-induced phosphorylation of CRYs may result in electrostatic repelling of the PHR and CCE domains of CRYs and open conformation of CRYs, which has been deduced from the conventional structure-function studies[35,45]. However, technical difficulties, including the intrinsically disordered nature of the CCE domain of CRY2, has prevented a direct test of conformational changes of the CRY2 holoprotein. Because COP1 interacts with CRY2 regardless of CRY2 phosphorylation, whereas LRBs primarily interacts with phosphorylated CRY2 (Fig. 3, Fig. 6e, f), we propose that the first type of conformational change is primarily responsible for the photoresponsive interaction of COP1 and CRY2, whereas the second type of conformational change of CRY2 is primarily responsible for the photoresponsive interaction of LRBs and CRY2. These hypotheses, however, remain to be further tested.

It is interesting that plants evolved with two distinct E3 ubiquitin ligases to regulate a CRY photoreceptor by different modes of interaction under different light conditions. In mammals, two related Cul1-based E3 ligases, SCF$^{FBXL3}$ and SCF$^{FBXL21}$, act to promote ubiquitination and degradation of CRYs in the nucleus and cytoplasm, respectively, to control the period of the circadian clock[8]. In Drosophila, Jetlag of a Cul1-based E3 ligase and Brwd3 of a Cul4-based E3 ligase regulate ubiquitination and degradation of dCRY-dependent circadian rhythm[9,10]. Cul4$^{Brwd3}$ mediates light-dependent ubiquitination and degradation of dCRY, whereas Cul1$^{Jetlag}$ interacts with dCRY to catalyze ubiquitination of the dCRY-interacting protein TIM[10,46]. Decreased TIM may enhance interaction of dCRY with its E3 ligases to accelerate its degradation in light. These studies demonstrate that multiple E3 ligases are needed to regulate the activity of a CRY to control the circadian clock. The results of this study argue that plant CRY2 is also controlled by two distinct E3 ligases, and that the delicate control of the abundance of CRYs is evolutionarily conserved for not only circadian rhythms in animals but also photomorphogenesis in plants. Our finding that Cul3$^{LRBs}$ or Cul4$^{COP1/SPAs}$ catalyze CRY2 ubiquitination in different light conditions, to promote rapid or prolonged degradation of CRY2, respectively, highlights the importance of photoreceptor turnover in the regulation of photosensitivity of plants. Altogether, these results appear to argue for the similar selection pressure to maintain multiple mechanisms regulating the abundance of CRY proteins during evolution of both animal and plant lineages. The complex mechanism regulating CRY2 turnover is likely evolved to optimize photomorphogenesis under natural light conditions. In this regard, it is particularly interesting that Cul3$^{LRBs}$ has been previously shown to be the E3 ligase of PIF3 and critical for the function of phyB and phytochrome-dependent photoresponses[30,31]. Our finding that Cul3$^{LRBs}$ also catalyzes blue light-dependent ubiquitination of CRY2 to regulate blue light-dependent photomorphogenesis argues for a previously unrecognized mechanism toward a better understanding of how plants respond to light in the nature white light conditions.

## Methods

**Plasmid construction.** All plasmids used in this study were generated by In-Fusion Cloning methods (https://www.takarabio.com/products/cloning/in-fusion-cloning). The sequences subcloned into plasmids were verified by Sanger sequencing. The mammalian cell expression vectors used in this study were made with pQCMV-Flag-GFP (Supplementary Fig. 4), pQCMV-GFP (Supplementary Fig. 4), and pCMV-Myc[20]. pQCMV-Flag-GFP vector was modified from the commercial plasmid pEGFP-N1 (Clontech), which allows the inserted gene expressed under

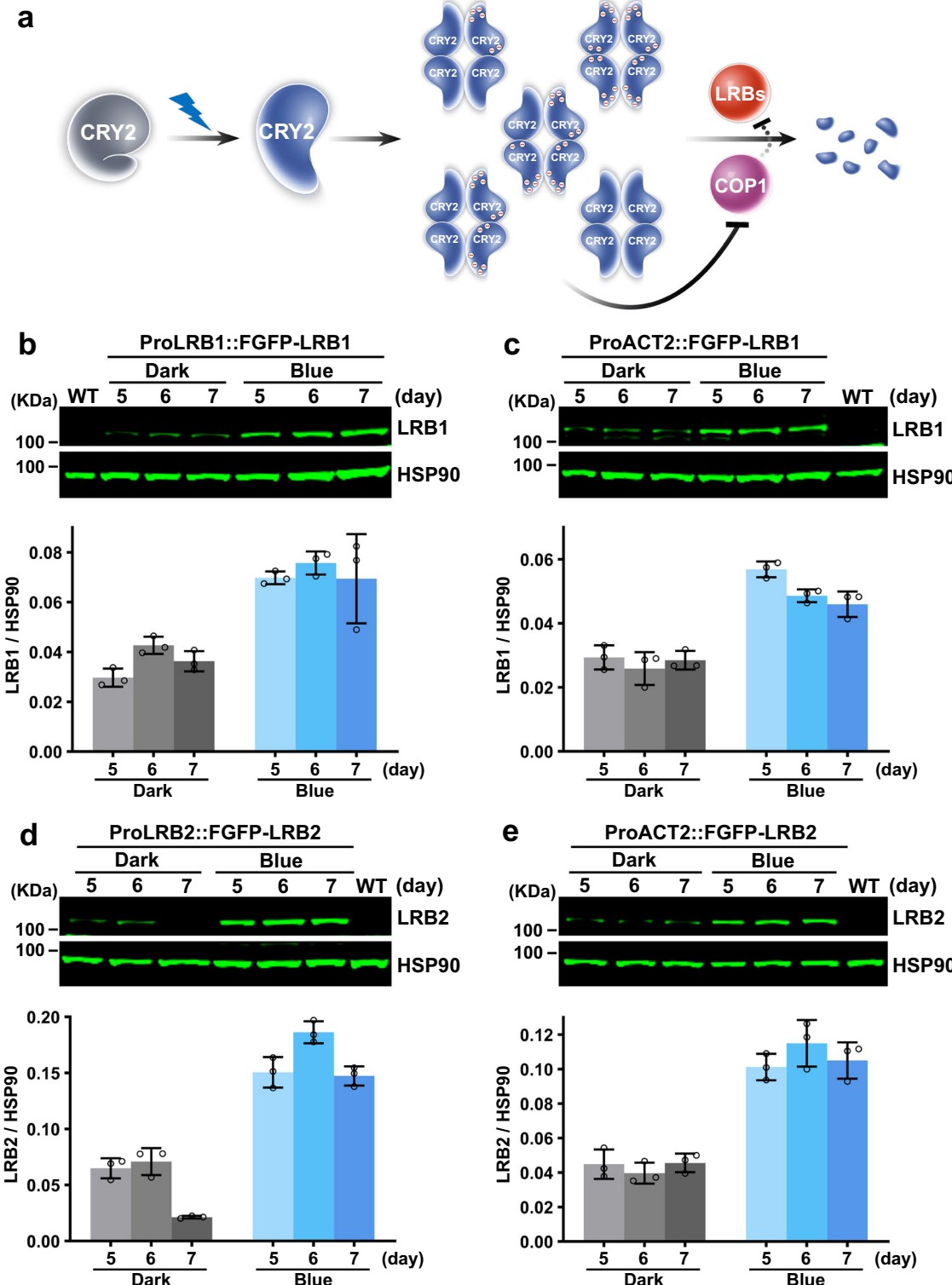

**Fig. 7 Blue light positively regulates LRBs protein abundance in plants. a** A hypothetic model depicting the degradation of CRY2 by LRBs and COP1 E3 ubiquitin ligases. CRY2 exists as inactive monomers in darkness (grey). Upon blue light exposure, CRY2 undergoes photoexcitation, oligomerization, and phosphorylation to become photoactive. Photoactivated CRY2 then were targeted by two distinct E3 ubiquitin ligases, LRBs, and COP1, for degradation. Minus sign in a circle, indicates negative charge; dashed line, indicates a hypothetical mechanism. **b–e** Representative immunoblots (top) and a quantitative analysis (bottom) showing the expression of recombinant LRB proteins in transgenic plants driven by native promoters (**b**, **d**) or the ACT2 constitutive promoter (**c**, **e**). Seedlings were grown in darkness or continuous blue light (50 μmol m$^{-2}$ s$^{-1}$) for 5, 6, or 7 days. Proteins were extracted and subjected to western blot analysis. LRBs and HSP90 were detected with anti-Flag and anti-HSP90 antibodies, respectively. HSP90 is used as a loading control. Data were shown as mean ± SD ($n = 3$ individual immunoblots). The above experiments were repeated at least three times with similar results.

control of the CMV promoter and fused with dual in-frame tags of Flag and GFP. pQCMV-GFP vector was further modified from pQCMV-Flag-GFP by removing the coding sequence of Flag tag. pCMV-Myc vector was described previously[20]. To generate plasmids expressing Flag-LRB1, Flag-LRB2, Flag-CRY2, and Flag-CRY2[D387A], the coding sequences (CDS) of all the above genes were amplified either from *Arabidopsis* cDNA or plasmids made before by PCR, the purified PCR products were then subcloned into SpeI/KpnI-digested pQCMV-Flag-GFP vector through in-fusion. *Myc-LRB1*, *Myc-LRB2*, *Myc-CRY2*, *Myc-SPA1*, and *Myc-COP1* plasmids were prepared by cloning the CDS of the genes into the BamHI site of pCMV-Myc vector. For plasmids expressing *HA-PPK1* and *HA-PPK1[D267N]*, the coding sequences of *PPK1* and *PPK1[D267N]* were PCR-amplified from plasmids made before by using the primers, of which a 2x HA coding sequence was attached to the 5' end of forward primer, and introduced into SpeI/KpnI-digested pQCMV-GFP vector by in-fusion. The gene-specific primers for constructs used for expressing recombinant proteins in HEK293T cells are listed in Supplementary Table 2.

The BiFC constructs were built with the vectors described previously[47]. The coding sequences of *LRB1* (AT2G46260), *LRB2* (AT3G61600), *CRY2* (AT1G04400), and *CRY2[D387A]* were PCR-amplified and cloned into pDONR/Zeo entry vector through BP reaction (Gateway™ BP Clonase™ II Enzyme mix, Cat. # 11789020, Invitrogen). The entry constructs were then introduced into the destination vector pX-nYFP (N-terminus of YFP fused to C-terminus of genes) or pcCFP-X (C-terminus of CFP fused to N-terminus of genes) by LR reaction (Gateway™ LR Clonase™ II Enzyme mix, Cat. # 11791020, Invitrogen). The gene-specific primers of Gateway cloning for constructs used for BiFC assys are listed in Supplementary Table 2.

The split-LUC constructs were made with the vectors described previously[48]. *CRY2* were fused to the C-terminus of cLUC (the C-terminus of Luciferase). *LRB1ΔBTB* were generated by seamlessly connecting two fragments, 1-426 bp and 637-1683 bp of *LRB1* coding sequence, with 15 bp overlaps, by infusion cloning method. *LRB2ΔBTB* were generated with two fragments, 1-432 bp and 751-1683 bp of *LRB2* coding sequence through the same method as making *LRB1ΔBTB*. *LRB1ΔBTB* and *LRB2ΔBTB* were then fused to the N-terminus of nLUC (the N-terminus of Luciferase) by in-fusion. The primers for constructs used for Split-LUC assays are listed in Supplementary Table 2.

pFGFP (Supplementary Fig. 5) and pDT1H (Supplementary Fig. 5) binary vectors were used for creating overexpression transgenic lines. The pFGFP binary vector was modified from pCambia3301[20], which allows the interested genes to express under Actin2 promoter and fuse with 2x Flag and GFP tags. The pDT1H binary vector, possessing two independent expression cassettes that allows two genes to assemble into one vector, was modified from previously published vector pDT1[49] by replacing the BAR gene with HPT gene to convert basta resistance to hygromycin resistance in plants. The CDS of *LRB1*, *LRB2*, *CRY2*, and *CRY2[P532L]* were cloned into BamHI-digested pFGFP vector to generate *FGFP-LRB1*, *FGFP-LRB2*, *FGFP-CRY2*, *FGFP-CRY2[P532L]* with 2x Flag and GFP tags fused to N-terminus of the genes. The native promoters of *LRB1* and *LRB2* were cloned into SacI/SpeI-digested *FGFP-LRB1* and *FGFP-LRB2* to generate *ProLRB1::FGFP-LRB1* and *ProLRB2::FGFP-LRB2*, respectively. The CDS regions of *LRB1*, *LRB2*, and *COP1* (AT2G32950) were cloned into the BamHI site of pDT1H vector to produce *Myc-LRB1*, *Myc-LRB2*, and *Myc-COP1* with 4x Myc tags fused to N-terminus of the genes. The primers for constructs used for plant transformation are listed in Supplementary Table 2.

### Plant materials and growth conditions.
All mutants used in this study are in the *Arabidopsis thaliana* Columbia ecotype background. The *cry1cry2* double mutant were described as previously[32]. The *lrb1lrb2-1lrb3* (*lrb1*, Salk_145146; *lrb2-1*, Salk_001013; *lrb3*, Salk_082868) and *lrb1lrb2-2lrb3* (*lrb2-2*, Salk_044446) triple mutants were gifts from Dr. Peter Quail and as described previously[30,31]. *cop1-4* and *cop1-6* are weak mutant alleles of *COP1* as previously described[50]. *cry1cry2lrb1lrb2-2lrb3* and *lrb1lrb2-2lrb3cop1-4* mutants were generated by crossing. The genotypes of *lrb123* mutants were verified by PCR using the primers listed in Supplementary Table 2. *cop1-4* mutant was verified by PCR using the primers listed in Supplementary Table 2, followed by Sanger sequencing to confirm the point mutation of *COP1* gene. *cry1cry2* mutant was verified by western blots using antibodies against CRY1 and CRY2 proteins.

All transgenic lines were generated via *Agrobacterium tumefaciens*–mediated floral-dip method[51]. The wild-type plants used for transformation in this study are *rdr6-11*, which suppresses gene silencing[52]. For in vivo ubiquitination study, *FGFP-CRY2* was introduced into *rdr6-11*, *lrb1lrb2-1lrb3*, *lrb1lrb2-2lrb3*, *cop1-4* and *cop1-6* background. The *FGFP-CRY2/Myc-LRB1* double-overexpression lines were prepared by introducing *FGFP-CRY2* into *Myc-LRB1/rdr6-11* plants. The transgenic T1 populations were screened on MS plates containing 25 mg/L Glufosinate-ammonium (cat # CP6420, Bomei Biotechnology) and 25 mg/L hygromycin (cat # 10843555001, Roche), and western blots were performed to confirm the expression of both proteins. The same method was used for generating *FGFP-CRY2/Myc-LRB2* and *FGFP-CRY2/Myc-COP1* double-overexpression lines. For experiments comparing the hypocotyl phenotype and protein degradation kinetics of FGFP-CRY2 and FGFP-CRY2[P532L], *FGFP-CRY2* and *FGFP-CRY2[P532L]* were introduced into *cry1cry2rdr6* background. The *cry1cry2rdr6* were generated by crossing *cry1-304*[32], *cry2-1*[12], and *rdr6-11*[52]. The transgenic lines were screened on MS plates with 25 mg/L Glufosinate-ammonium, and lines with similar protein

expression level of FGFP-CRY2 and FGFP-CRY2[P532L] were used for analysis. For *lrb123* mutant blue-light hypersensitivity phenotype rescue experiments, *FGFP-LRB2* and *Myc-LRB2* were transformed into *lrb1lrb2-2lrb3* background.

For routine maintenance, *Arabidopsis thaliana* were grown under long day conditions (16 h light / 8 h dark) at 22 °C. For hypocotyl phenotype analysis, seedlings were grown on MS plates with 3% sucrose at 20–22 °C for 6 days under different light conditions. Light-emitting diode (LED) was used to obtain monochromatic light (blue light, peak 465 nm, half-bandwidth of 25 nm; red light, peak 660 nm, half-bandwidth of 20 nm; far-red light, peak 735 nm, half-bandwidth of 21 nm). For endogenous CRY2 degradation analysis in WT, *lrb123*, *cop1* and *lrb123cop1* mutants, seedlings were grown in darkness on MS plates with 3% sucrose for 7 days, then subjected to 30 µmol m$^{-2}$ s$^{-1}$ blue light for the indicated time. For FGFP-CRY2 and FGFP-CRY2[P532L] degradation analysis, seedlings were grown in darkness on MS plates with 3% sucrose for 7 days, then subjected to 100 µmol m$^{-2}$ s$^{-1}$ blue light for the indicated time. For immunoprecipitation of polyubiquitinated proteins, 7-day-old etiolated seedlings grown on MS medium containing 3% sucrose were treated with 50 µM MG132 (Cat # S2619, Selleck) in liquid MS in the dark overnight with gentle shaking, and then moved to 30 µmol m$^{-2}$ s$^{-1}$ blue light for 5, 10, and 15 min before harvest.

### Blue-light-induced CRY2 degradation assays.
For endogenous CRY2 degradation analysis, seedlings were grown in the dark for 7 days and then treated with 30 µmol m$^{-2}$ s$^{-1}$ of blue light for indicated time. For FGFP-CRY2 and FGFP-CRY2[P532L] degradation analysis, seedlings were treated with 100 µmol m$^{-2}$ s$^{-1}$ of blue light. Tissues were homogenized by TissueLyser (QIAGEN). Proteins were extracted in equal tissue volume of protein extraction buffer [120 mM Tris pH 6.8, 100 mM EDTA, 4% w/v SDS, 10% v/v beta-mercaptoethanol, 5% v/v Glycerol, 0.05% w/v Bromophenol blue][53], heated at 100 °C for 8 min, centrifuged at 16000 rcf for 10 min, and analyzed by western blot. Proteins were separated in 10% SDS-PAGE, and transferred to nitrocellulose membranes (Pall Life Sciences).

For immunoblot signals detected by enhance conventional chemiluminescent (ECL) method, membranes were blocked with 5% non-fat milk in PBST for 1 h, blotted with anti-CRY2 primary antibody (1:3000, homemade, produced in rabbit[12]) in PBST for 1.5 h, washed 8 min x 3 times with PBST, blotted with anti-Rabbit-HRP (1:10000, cat # 31460, ThermoFisher) secondary antibody in PBST for 1.5 h, washed 8 min x 3 times with PBST and then incubated with ECL solution for X-ray film development. The membranes were then stripped with stripping buffer [0.2 M Glycine, pH 2.5] and re-probed with anti-HSP82 (1:10000, cat # AbM51099-31-PU, Beijing Protein Innovation) primary antibody and anti-Mouse-HRP (1:10000, cat # 31430, ThermoFisher) secondary antibody. Anti-HSP82 antibody from *Oryza sativa* can also recognize *Arabidopsis* HSP90 protein[54]. Three independent blots were performed in parallel for quantification analysis. The quantification of protein intensity of ECL immunoblots was performed by ImageJ.

For immunoblot signals detected by Odyssey CLx Infrared Imaging System (Li-COR), membranes were blocked with 0.5% casein in PBS for 1 h, blotted with anti-CRY2 (1:3000) and anti-HSP82 (1:10000) mixed primary antibodies in PBST with 0.5% casein for 1.5 h, washed 8 min x 3 times with PBST, blotted with Donkey anti-rabbit 790 (1:15000, cat # A11374, ThermoFisher) and Donkey anti-mouse 680 (1:15000, cat # A10038, ThermoFisher) mixed secondary antibodies in PBST with 0.5% casein for 1.5 h, washed 8 min x 3 times with PBST and then signals were captured with Odyssey CLx by Image Studio Lite software. Three independent blots were performed in parallel for quantification analysis. Quantification of signals were processed with Image Studio Lite software. CRY2 degradation curves were indicated by CRY2 (B/D) ratio, calculated as CRY2 (B/D) = [CRY2/HSP90]$^{blue}$ / [CRY2/HSP90]$^{dark}$, and analyzed with one phase decay of nonlinear regression.

### Protein expression and co-immunoprecipitation in HEK293T cells.
Human embryonic kidney (HEK) 293T cells were cultured in Dulbecco's modified Eagle's medium (DMEM) supplemented with 10% (v/v) FBS, 100 mg/L streptomycin and 100 IU penicillin, in humidified 5% (v/v) CO$_2$ air at 37 °C. Cells were seeded at a density of approximately $8 \times 10^5$ cells/6-cm plate and the transfection were carried out with a calcium phosphate precipitation protocol. Briefly, different combinations of plasmid DNA (2~5 µg / construct) were mixed with 30 µl 2.5 M CaCl$_2$ and ddH$_2$O to a total volume of 300 µl, then 300 µl of 2x HeBS [250 mM NaCl, 10 mM KCl, 1.5 mM Na$_2$HPO$_4$, 12 mM Dextrose and 50 mM HEPES, adjust the pH of the final solution to 7.05] was added drop by drop with vortex, and kept at room temperature for 5 min before applying to cells. The media were aspirated from each plate, DNA mixtures were gently added into plates and the plates were rotated gently to allow the mixtures to coat the entire plate. 3 ml of fresh media containing 25 µM chloriquine (cat # C6628, Sigma) were added to each plate and the plates were kept in the CO$_2$ incubator overnight. The next day, the media were changed with 3 ml of fresh media without chloriquine. 36–48 h after transfection, the cells were subjected to blue light treatment for indicated time, washed twice with cold PBS buffer and then harvested in liquid nitrogen for co-immunoprecipitation.

Cells transfected with different plasmid DNA were lysed in 800 µl 1% Brij buffer [1% Brij-35, 50 mM Tris-HCl pH 8.0, 150 mM NaCl, 1 mM EDTA, 1x Protease inhibitor cocktail, and 1x phosphatase inhibitor PhosSTOP] with rotating at 4 °C for 20 min. Cell lysates were centrifuged at 16,000 rcf for 10 min at 4 °C, and the supernatants were incubated with 20 µl Anti-FLAG M2 affinity gel (Cat. # F2426, Sigma) at 4 °C for 2 h with rotation. Beads were washed with 1% Brij buffer for 5

times. Proteins were competed from the beads with 35 µl 3xFlag peptide solution [500 ng/µl in 1% Brij buffer] for 30 min at room temperature with mixing. Elution was transferred to a new tube, mixed with 4XSDS sample buffer, denatured at 100 °C for 4 min and subjected to western blot analysis. Western blots were performed as described above. Primary antibodies used here were anti-Flag (1:1500, cat # F1804, Sigma), anti-Myc (1:5000, cat # 05-724, Millipore), and anti-HA (1:5000, cat # 12013819001, Roche), and secondary antibodies used were anti-Mouse-HRP (1:10000, cat # 31430, ThermoFisher) and anti-Rabbit-HRP (1:10000, cat # 31460, ThermoFisher).

**Bimolecular fluorescence complementation (BiFC) assay**. BiFC assays in *Arabidopsis* plants were performed as previously described methods with minor modifications[37]. Briefly, *Agrobacteria* AGL0 transformed with BiFC plasmids were grown in LB medium with 40 µM Acetosyringone and 1% Glucose in 28 °C shaker overnight. *Agrobacteria* were centrifuged at 5000 rcf for 10 min, washed once with washing buffer [10 mM MgCl$_2$, 100 µM Acetosyringone] and resuspended in infiltration buffer [1/4 MS pH 6.0, 1% sucrose, 100 µM Acetosyringone, and 0.01% Silwet L-77] to 0.5 of OD600. The *agrobacteria* were infiltrated into 3–4 weeks old *Arabidopsis* leaves with syringe. Each BiFC assay was performed with at least 3 independent plants with 2–3 leaves infiltrated for each. Plant leaves were dried before kept in the dark for 24 h, and then moved back to long day conditions (16 h light/8 h dark) to grow for 2 more days. The infiltrated leaf samples were analyzed under a Leica TCS SP8X confocal microscope. Hoechst 33342 was used for nuclei staining. The quantification of fluorescence intensity was performed by ImageJ. Briefly, the images of GFP and Hoechst 33342 channels from the same field were stacked first, then the regions of nuclei were selected on the image of Hoechst 33342 channel. The integrated intensity of the selected regions over the entire stack were measured. The BiFC ratio of the selected regions were calculated as [GFP intensity]$^{nuclei}$ / [Hoechst 33342 intensity]$^{nuclei}$ for each image. Four to nine images (~40–100 nuclei in total) for each BiFC assay were used for quantification and the BiFC ratios were presented as mean ± SD (standard deviation).

**Split-luciferase (Split-LUC) assay**. Split-LUC plasmids were transformed into *Agrobacteria* AGL0. Each *Agrobacteria* with respective plasmid was grown in LB medium with 40 µM Acetosyringone and 1% Glucose in 28 °C shaker overnight. *Agrobacteria* were centrifuged at 5000 rcf for 10 min, washed once with washing buffer [10 mM MgCl$_2$, 100 µM Acetosyringone] and resuspended in infiltration buffer [1/4 MS pH 6.0, 1% sucrose, 100 µM Acetosyringone, and 0.01% Silwet L-77] to 0.5 of OD600. *Agrobacteria* with the nLUC- or cLUC-fusion proteins were mixed at a ratio of 1:1 and were infiltrated into *N. benthamiana* leaves with syringe. Each split-LUC assay was performed with at least 3 leaves. The bacteria culture remained on the surface of plant leaves were dried before the plants were kept in the dark for 24 h, and then moved back to long day conditions (16 h light/8 h dark) to grow for 2 more days. Before the analysis, 1 mM D-luciferin with 0.01% Triton X-100 were sprayed on the leaves, and the leaves were kept in the dark for 20 min before signal detecting with Tanon-5200. The quantification of luminescence intensity was performed by ImageJ.

**Immunoprecipitation of ubiquitinated proteins in *Arabidopsis***. 7-day-old etiolated seedlings were treated with 50 µM MG132 in liquid MS in the dark overnight with gentle shaking, and then treated with 30 µmol m$^{-2}$ s$^{-1}$ blue light for 5, 10, and 15 min before harvest. For each immunoprecipitation assay, about 3 g of seedlings were used. Seedlings were ground in liquid nitrogen and homogenized in 1.5x tissue volume of IP buffer [50 mM Tris-HCl pH 7.5, 150 mM NaCl, 1 mM EDTA, 1% Triton X-100, 20 mM NaF, and 1x Protease inhibitor cocktail] at 4 °C for 20 min with mixing. Lysates were centrifuged at 16,000 rcf for 15 min at 4 °C. A total of 100 µl of supernatant were saved as inputs.

For total ubiquitinated protein purification, the supernatant for each sample was incubated with 30 µl Agarose-TUBE2 beads (cat # UM402M, LifeSensors) for 2 h at 4 °C with rotating. Then, beads were pelleted and washed four times with ice-cold IP buffer (without inhibitors). Total ubiquitinated proteins were eluted with 2XSDS sample buffer, denatured at 100 °C for 4 min and subjected to western blot analysis. Anti-ubiquitin antibody (α-Ubq, cat # 14-6078-80, Thermofisher) was used to detect total ubiquitinated proteins, and homemade anti-CRY2 antibody was used to detect CRY2 protein.

For FGFP-CRY2 protein purification, the cleared plant lysate were incubated with 50 µl GFP-trap agarose beads (homemade or cat # gta-20, Chromotek) and rotated at 4 °C for 2 h. Beads were pelleted and washed four times with IP buffer (without inhibitors). Proteins were eluted with 2XSDS sample buffer by heating at 100 °C for 4 min and subjected to western blot analysis. Anti-Flag antibody was used to detect FGFP-CRY2 (IP) fusion protein, and anti-ubiquitin antibody was used to detect ubiquitinated CRY2.

**Statistics and reproducibility**. For Fig. 1b, n = 16 for WT (dark), n = 15 for WT (red), n = 17 for WT (FR), n = 14 for WT (LD), n = 17 for WT (SD), n = 22 for *cry1cry2* (dark), n = 17 for *cry1cry2* (red), n = 22 for *cry1cry2* (FR), n = 22 for *cry1cry2* (LD), n = 22 for *cry1cry2* (SD), n = 20 for *lrb1lrb2-2lrb3* (dark), n = 25 for *lrb1lrb2-2lrb3* (red), n = 25 for *lrb1lrb2-2lrb3* (FR), n = 19 for *lrb1lrb2-2lrb3* (LD),

n = 20 for *lrb1lrb2-2lrb3* (SD), n = 18 for *cry1cry2lrb1lrb2-2lrb3* (dark), n = 25 for *cry1cry2lrb1lrb2-2lrb3* (red), n = 25 for *cry1cry2lrb1lrb2-2lrb3* (FR), n = 17 for *cry1cry2lrb1lrb2-2lrb3* (LD), n = 18 for *cry1cry2lrb1lrb2-2lrb3* (SD).

For Fig. 1f, n = 23 for WT (dark), n = 23 for *cry1cry2* (dark), n = 20 for *lrb1lrb2-2lrb3* (dark), n = 20 for *cry1cry2lrb1lrb2-2lrb3* (dark), n = 20 for WT (blue), n = 25 for *cry1cry2* (blue), n = 20 for *lrb1lrb2-2lrb3* (blue), n = 25 for *cry1cry2lrb1lrb2-2lrb3* (blue).

For Fig. 1h, n = 21 for WT (dark), n = 24 for *lrb1lrb2-2lrb3* (dark), n = 20 for *cop1-4* (dark), n = 25 for *lrb1lrb2-2lrb3cop1-4* (dark), n = 26 for WT (blue), n = 15 for *lrb1lrb2-2lrb3* (blue), n = 23 for *cop1-4* (blue), n = 19 for *lrb1lrb2-2lrb3cop1-4* (blue).

For Fig. 6b, n = 25 for all fluence rates, except n = 21 for WT (30 µmol m$^{-2}$ s$^{-1}$), n = 24 for *cry1cry2* (0 µmol m$^{-2}$ s$^{-1}$), n = 21 for FGFP-CRY2 (30 µmol m$^{-2}$ s$^{-1}$).

For Supplementary Fig. 1d, e, n = 18 for WT, n = 15 for *cry1cry2*, n = 17 for *lrb1lrb2-2lrb3*, n = 15 for *cry1cry2lrb1lrb2-2lrb3*.

For Supplementary Fig. 2b, n = 25 for all, except n = 24 for *Myc-LRB2/lrb1lrb2-1lrb3* #61.

For Supplementary Fig. 2d, n = 20 for WT (Dark), n = 20 for *cry1cry2* (Dark), n = 25 for *lrb1lrb2-2lrb3* (Dark), n = 21 for FGFP-LRB2/*lrb1lrb2-2lrb3* #5 (Dark), n = 20 for FGFP-LRB2/*lrb1lrb2-2lrb3* #8 (Dark), n = 20 for FGFP-LRB2/*lrb1lrb2-2lrb3* #17 (Dark), n = 25 for all genotypes in blue.

**Reporting summary**. Further information on research design is available in the Nature Research Reporting Summary linked to this article.

## Data availability
The source data for Figs. 1–7, Supplementary Figs. 1–3 are provided with this paper as a Source Data file. Other data and materials of this study are available from the corresponding author upon reasonable request.

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

## Acknowledgements

The authors thank Dr. Peter Quail for providing the *lrb123* mutant alleles and Dr. Hongquan Yang for providing the split-LUC vectors. Works in the authors' laboratories are supported in part by the National Natural Science Foundation of China (31970265 to Q.W.), Natural Science Foundation of Fujian Province (2019J06014 to Q.W.), FAFU-ICE fund (KXGH17011 to Q.W.) and the National Institutes of Health (GM56265 to C.L.). The UCLA-FAFU Joint Research Center on Plant Proteomics provided the institutional supports.

## Author contributions

Y.C. and X.H. conducted most of the experiments. S.L. and T.S. performed some biochemical experiments in HEK293T. H.H. and H.R. prepared some genetic materials. S.L. and Z.G. helped with BiFC experiments. X.W. and D.L. provided critical feedback. J.W. provided mass-spectrometry supports. C.L. and Q.W. conceived the project, designed the experiments, and wrote the paper.

## Competing interests

The authors declare no competing interests.
