## [Peer Review File · Nature Communications]

REVIEWER COMMENTS

Reviewer #1 (Remarks to the Author):

CRY2 is a light-labile photoreceptor that is rapidly degraded in blue light. Up to now, only COP1/SPA had been identified as a responsible E3 ubiquitin ligase. Here, the authors report an additional E3 ligase, the LRB family, and show that both COP1/SPA and LRBs are responsible for CRY2 degradation. LRBs were previously identified as an E3 ligase on PIFs and phyB. This manuscript provides very novel data in two-ways: it finally clarifies the E3 ligases that cause CRY2 degradation that have been long sought for; second, it demonstrates that LRBs are involved in both red light and blue light signaling.

The results presented on CRY2 polyubiquitination and degradation are convincing and support the authors' conclusions. The manuscript is well-written. The results will be of great interest to a larger readership, also beyond the photobiology community.

I have two major comments:

1. Most protein-protein interactions are shown in HEK cell expression systems. I would appreciate additional evidence that LRBs interact with CRY2 *in vivo*. BiFC is not very stringent (also, more negative controls using "empty vector" or unrelated fusion proteins should be included). The authors generated LRB CRY2 double transformants; hence, have they attempted *in vivo* coIPs?
2. The authors show nicely that LRBs cause CRY2 degradation transiently after transfer of plants from darkness to light. It would be good if they could discuss why LRBs act transiently while COP1 acts short-term and long-term. Are LRBs instable in blue light? Does expression change?

Minor comments:

Fig 3a and b: what is the difference between the experiments shown in a and b? They appear identical, but PPK1 was pulled down by FLAG-LRB2 only in blue light (b) or in darkness and blue light (a).

Fig. 5. Please indicate more clearly which duration of blue light irradiation (5, 10, 15 min?) was used in which subfigure. It is not clear from the legend.

Fig. 4: Why is BiFC fluorescence intensity normalized to a nuclear stain? Normalizing to a co-expressed fluorescent marker would be useful to normalize for transfection efficiency. But how would normalizing for nuclear stain improve the data?

Please provide more detailed information on the plasmid constructs used. The descriptions are very cryptic, e.g. vector backbones (citation possible?), primer sequences, etc.

Reviewer #2 (Remarks to the Author):

cry2 is a blue light photoreceptor crucial for photoperiodic flowering and light-mediated inhibition of cell elongation. Upon blue light activation, cry2 is phosphorylated by PPK kinases, followed by ubiquitination and proteasomal degradation. Previous work by the Lin lab has identified COP1/SPA as a E3 ubiquitin ligase complex that targets cry2 for ubiquitination and proteasomal degradation. In addition, cry2 is known as a negative regulator of COP1 activity, resulting in the stabilization of

its substrates.

As blue-light-dependent cry2 degradation was found to be only partially impaired in cop1 null mutants, the existence of additional E3 ubiquitin ligases targeting cry2 has been postulated. In this work, Yadi Chen et al describe the discovery of the CUL3-LRBs E3 ubiquitin ligases as key enzymes ubiquitinating cry2 and thereby targeting cry2 for proteasomal degradation. Interestingly, the authors find that COP1 determines cry2 protein abundance under steady state conditions during continuous blue light exposure (and not LRBs), whereas LRBs are responsible for the rapid blue light-induced cry2 degradation (and not COP1). The complementarity of the two distinct ubiquitin ligases - one for the rapid ubiquitination and degradation of cry2, the other one for the slow or prolonged ubiquitination and degradation of cry2 - is very clearly illustrated by the cry2 stability in lrb1 lrb2 lrb3 cop1 quadruple mutants (Fig. 2e). This work provides very interesting and novel findings that significantly further our understanding of cry2 regulation. The manuscript is very well written and presented, with high quality experimental data supporting the conclusions.

A few points to be addressed:

1. cry2 uses a VP-motif to interact with COP1, as other COP1 substrates do. At present, the VP motif in photoreceptors has been mainly discussed as mimicking substrate interaction motifs, thereby resulting in COP1 target proteins to get stabilized (i.e. by competing off the COP1 substrates) (Ponnu et al., 2019; Lau et al., 2019). It would be interesting to discuss shortly that cry2 may function at the same time by inhibiting COP1, and as a COP1 substrate. It is not very clear how cry2 inhibits COP1 when it is ubiquitinated and degraded as any other substrate. It also is surprising that COP1 seems involved in late degradation of cry2. Is it that cry2 binds to COP1 inactivating it, but then only slowly becomes a substrate for COP1 ubiquitination? Any such mechanism known for any E3 ligase (different kinetics of ubiquitination after binding to an E3 ligase)? Could COP1 binding even protect a subfraction of cry2 from being targeted by LRBs? It may be interesting to shortly discuss.

cry2P532L is thought to block interaction with COP1, and not LRBs (e.g. Fig. 6 e, f). It is not clear to me why degradation of cry2P532L then rather mimicks lrb mutants than cop1 mutants (compare Fig. 6c with Fig. 2e). P532L seems stabilized early and not late. Any explanation for that?

2. Fig. 5d: long-exposure suggest that cry2 is even targeted in darkness by LRBs and COP1. It is less clear if that is reflected by cry2 protein levels in darkness (i.e. less cry2 in WT than in lrb1 lrb2 lrb3, cop1, and lrb1 lrb2 lrb3 cop1). Please mention and discuss shortly.

3. cry2 is introduced as an important regulator of photoperiodic flowering (line 38). Is anything known about lrb and flowering? If data available, it would be great to add. If not, I think it would be good to shortly mention in the discussion that it will need to be investigated whether LRBs "only" with role for hypocotyl growth inhibition or also flowering time regulation; or whether such an involvement is not expected (at least not due to directly targeting cry2 for degradation).

Minor:

- lines 34, 249: protein name "Jetlag", not "Jetleg"
- provide full name for TIM protein

Authors' responses:

MS ID#: NCOMMS-20-45996

Title: Regulation of Arabidopsis photoreceptor CRY2 by two distinct E3 ubiquitin ligases

By: Yadi Chen, Xiaohua Hu, Siyuan Liu, Tiantian Su, Hsiao-chi Huang, Huibo Ren, Zhensheng Gao, Xu Wang, Deshu Lin, James A. Wohlschlegel, Qin Wang and Chentao Lin

Reviewer #1:

Major comments:

1. Most protein-protein interactions are shown in HEK cell expression systems. I would appreciate additional evidence that LRBs interact with CRY2 *in vivo*. BiFC is not very stringent (also, more negative controls using “empty vector” or unrelated fusion proteins should be included). The authors generated LRB CRY2 double transformants; hence, have they attempted *in vivo* colIPs?

Response:

We thank the reviewer for this insightful comment. We added results of two new *in vivo* experiments by split-LUC (Fig. 4b-c) and co-IP (Fig. 4d-e). The new results further support the CRY2/LRBs interaction argument.

2. The authors show nicely that LRBs cause CRY2 degradation transiently after transfer of plants from darkness to light. It would be good if they could discuss why LRBs act transiently while COP1 acts short-term and long-term. Are LRBs instable in blue light? Does expression change?

Response:

This is a very insightful comment that would significantly improve this paper. We added two discussions specifically to this issue (page 5, line 148-154; page 11, line 315-333), and we also added new data to show the photoresponsive LRB protein expression (Fig. 7b-e).

Minor comments:

Fig 3a and b: what is the difference between the experiments shown in a and b? They appear identical, but PPK1 was pulled down by FLAG-LRB2 only in blue light (b) or in darkness and blue light (a).

Response:

The experimental conditions for Fig. 3a and 3b are the same, except that 3a is for LRB1 and 3b is for LRB2. In the revision, we replaced PPK1 signal of Fig. 3b with a stronger exposure of the original experiment to minimize potential misleading impressions. It is possible that the affinity of PPK1 to CRY2 may be different in the presence of LRB1 or LRB2, but we do not wish to make this conclusion at present, and it seems irrelevant to our main conclusion.

Fig. 5. Please indicate more clearly which duration of blue light irradiation (5, 10, 15 min?) was used in which subfigure. It is not clear from the legend.

Response:

We thank the reviewer for pointing out this oversight of ours. It is clarified in legend for the revision. We exposed seedlings with three different time treatment and mixed the seedling samples to see the overall light response.

Fig. 4: Why is BiFC fluorescence intensity normalized to a nuclear stain? Normalizing to a co-expressed fluorescent marker would be useful to normalize for transfection efficiency. But how would normalizing for nuclear stain improve the data?

Response:

Our purpose is to normalize the background fluorescence, such as unknown source of autofluorescence, for BiFC signals. We tend to think that, as long as the transfection conditions are carefully controlled, the major artifacts of BiFC may be from the background autofluorescence, but not the common differential levels of protein expression encountered in the conventional fluorescence experiment. We agree that a better way is to include independent assays, so we added independent assays as the reviewer pointed out in the major comment (Fig. 4b-e).

Please provide more detailed information on the plasmid constructs used. The descriptions are very cryptic, e.g. vector backbones (citation possible?), primer sequences, etc.

Response:

We now include two new supplemental figures to show the maps of the vectors used in this study (Supplementary Fig. 4, 5) and the primers used for plasmid construction are in Supplementary Table 2.

Reviewer #2:

Major comments:

1. ...It would be interesting to discuss shortly that cry2 may function at the same time by inhibiting COP1, and as a COP1 substrate....Is it that cry2 binds to COP1 inactivating it, but then only slowly becomes a substrate for COP1 ubiquitination? Any such mechanism known for any E3 ligase (different kinetics of ubiquitination after binding to an E3 ligase)? Could COP1 binding even protect a subfraction of cry2 from being targeted by LRBs?cry2P532L is thought to block interaction with COP1, and not LRBs (e.g. Fig. 6 e, f). It is not clear to me why degradation of cry2P532L then rather mimicks lrb mutants than cop1 mutants (compare Fig. 6c with Fig. 2e). P532L seems stabilized early and not late. Any explanation for that?

Response:

We thank the reviewer for these insightful comments. We added additional discussion to specifically address the reviewer's questions (page 5, line 148-154; page 10, line 291-305; page 11-12, line 315-352).

We agree with the reviewer that different role of COP1 and LRB in CRY2 signaling is likely one of the explanations. We are not aware of any other E3 ligase acts this way (substrate inhibits its own E3 ligase). However, our new results (Fig. 7b-e) included in the revision is consistent with this comment by the reviewer, and it suggests a possible mechanism. Thus, we tempt to make this speculative proposition that COP1 may suppress LRB by facilitating its degradation in the dark, similar to other proteins, such as HY5. It is also possible that COP1 binds subclass of CRY2 with different consequences, and we emphasize in the revision the difference between COP1 and LRB with respect to their interaction with different structural elements of CRY2.

We added new Fig. 6g-j, which may partially explain the unusual behavior of CRY2^{P532L}. Namely, the VP mutation may alter the photochemistry of this mutant to change its degradation behavior. Although the exact photochemical mechanism of this is out of the scope of this study, we emphasize this unexpected results and possible explanations in the revision (page 10, line 291-305).

2. Fig. 5d: long-exposure suggest that cry2 is even targeted in darkness by LRBs and COP1. It is less clear if that is reflected by cry2 protein levels in darkness (i.e. less cry2 in WT than in lrb1 lrb2 lrb3, cop1, and lrb1 lrb2 lrb3 cop1). Please mention and discuss shortly.

Response:

The reviewer raised an excellent question. We now include a short discussion about this complex result (page 8, line 237-241).

3. cry2 is introduced as an important regulator of photoperiodic flowering (line 38). Is anything known about lrb5 and flowering? If data available, it would be great to add. If not, I think it would be good to shortly mention in the discussion that it will need to be investigated whether LRBs "only" with role for hypocotyl growth inhibition or also flowering time regulation; or whether such an involvement is not expected (at least not due to directly targeting cry2 for degradation).

Response:

The reviewer raised a relevant and excellent question. We include the flowering data in the revision (Supplementary Fig. 1) and a brief discussion (page 3, line 90-96).

Minor:

- lines 34, 249: protein name "Jetlag", not "Jetleg"
- provide full name for TIM protein

Response:

We made these corrections in the revision.

REVIEWERS' COMMENTS

Reviewer #1 (Remarks to the Author):

The authors did an excellent job in revising the manuscript. Hence, all my comments were addressed very well.

Reviewer #2 (Remarks to the Author):

The authors satisfactorily responded to all my requests and comments.